# Sodium-Glucose Cotransporter 2 (SGLT2) Inhibitors: Harms or Unexpected Benefits?

**DOI:** 10.3390/medicina59040742

**Published:** 2023-04-10

**Authors:** Munteanu Madalina Andreea, Swarnkar Surabhi, Popescu Razvan-Ionut, Ciobotaru Lucia, Nicolae Camelia, Tufanoiu Emil, Nanea Ioan Tiberiu

**Affiliations:** 1Department of Cardiology, University of Medicine and Pharmacy “Carol Davila”, 050474 Bucharest, Romania; 2“Theodor Burghele” Clinical Hospital, 050653 Bucharest, Romania; 3Department of Cardiovascular Science, University Medical Center Gottingen, 37075 Gottingen, Germany; 4Department of Urology, University of Medicine and Pharmacy “Carol Davila”, 050474 Bucharest, Romania; 5Department of Nephrology, University of Medicine and Pharmacy “Carol Davila”, 050474 Bucharest, Romania; 6Department of Neurology, University of Medicine and Pharmacy “Carol Davila”, 050474 Bucharest, Romania; 7Fundeni Clinical Institute, 022328 Bucharest, Romania

**Keywords:** heart failure, SGLT2 inhibitors, type 2 diabetes, cardiovascular effects

## Abstract

There is a need for innovative pharmaceutical intervention in light of the increasing prevalence of metabolic disease and cardiovascular disease. The kidneys’ sodium-glucose cotransporter 2 inhibitors (SGLT2) receptors are targeted to reduce glucose reabsorption by SGLT2. Patients with type 2 diabetes mellitus (T2DM) benefit the most from reduced blood glucose levels, although this is just one of the numerous physiological consequences. To establish existing understanding and possible advantages and risks for SGLT2 inhibitors in clinical practice, this article will explore the influence of SGLT2 inhibitors on six major organ systems. In addition, this literature review will discuss the benefits and potential drawbacks of SGLT2 inhibitors on various organ systems and their potential application in therapeutic settings.

## 1. Introduction

A recently developed family of drugs known as sodium-glucose cotransporter-2 (SGLT2) inhibitors lowers serum glucose by preventing the absorption of glucose by the tubules and increasing the excretion of glucose through the urine [1].

The S2 and S3 parts of the proximal tubules (along with the intestine) are the primary sites for type 1 and type 2 sodium-glucose cotransporters, which reabsorb a significant amount of filtrated urine glucose [2]. Around 80–90% of the glucose reabsorption process is carried out by SGLT 2 and the remaining 10–20% by SGLT 1 [3].

Six oral SGLT2 inhibitors are currently approved for the treatment of type 2 diabetes mellitus (T2DM) by the US Food and Drug Administration (FDA) together with the European Medicines Agency (EMA): canagliflozin (CANA), empagliflozin (EMPA), dapagliflozin (DAPA), ertugliflozin (ERTU), bexagliflozin, and sotagliflozin.

The European Society of Cardiology’s (ESC) recommendations for the year 2021 state that individuals with heart failure and reduced ejection fraction (HFrEF) are eligible for first-line treatment with SGLT2 inhibitors like dapagliflozin and empagliflozin. Regardless of the existence of diabetes, these medications should be given together with other first-line recommendations unless they are contraindicated or poorly tolerated (class I). In addition, patients with T2DM who are at risk for cardiovascular events should utilize canagliflozin, empagliflozin, dapaglifozin, sotagliflozin, and ertugliflozin. This is done to reduce the number of hospitalisations caused by heart failure (HF), major cardiovascular events (MACE), end-stage renal disease (ESRD), and CV deaths (also class I) [4]. Table 1 shortly illustrates the pharmacological properties of the SGLT2 inhibitors that are currently on the market.

The strong relationship between diabetes and HF results from the deleterious influence of key pathogenic variables, including chronic glucotoxicity, lipotoxicity, and altered insulin signalling. Decades of research have shown that diabetes causes detrimental comorbidities and consequences. Chronic hyperglycemia induces alterations in the cytoarchitecture of the Langerhans’ islets, including α cell hyperplasia, pancreatic beta cell dedifferentiation into glucagon-producing cells, and loss of paracrine and endocrine control due to cell mass loss [5].

Oxidative stress, increased production of advanced glycation end products, different intracellular calcium handling, inflammation, and endothelial dysfunction cause structural and functional abnormalities in the myocardium [6]. A 1% increase in glycosylated hemoglobin A1c (HbA1c) is associated with a 25% increased risk for cardiovascular events or death in patients with T2DM, according to the results of the Candesartan in Heart Failure: Assessment of Mortality and Morbidity (CHARM) research [7]. HF is prevalent among T2DM patients, with an estimated prevalence of >64 million new cases [8]. This emphasizes how crucial it is for T2DM patients to maintain strict glycemic control and cardiovascular treatment. The effects of SGLT2 inhibitors on glucose have been extensively studied, regardless of whether these medications are taken alone or in combination with other glucose-modulating medications (such as metformin, glucagon-like polypeptide 1 (GLP-1) agonists, dipeptidyl peptidase 4 (DPP-4) inhibitors, or insulin). They have been demonstrated to significantly aid the preservation of ideal glucose homeostasis [9,10,11].

This literature review will explore the advantages and potential risks of SGLT2 inhibitors on various organ systems, as well as their prospective use in therapeutic situations.

## 2. Materials and Methods

This literature narrative review investigates the effects of SGLT2 inhibitors, both positively and negatively, on six essential organs and systems. These systems consist of the kidneys, the liver, the pancreas, the nervous system, the pulmonary system, and the cardiovascular system. This article does not feature any new investigations; instead, it is based on research that has already been done. A review of the literature found in Scopus and PubMed was done between January 2016 and January 2023. Several combinations of the keywords, such as SGLT2 inhibitor, type 2 diabetes mellitus, cardiovascular effects, heart failure, and renal effects, were employed. Observational research, review articles, systematic reviews, and randomized controlled trials were also checked.

## 3. Results

### 3.1. Benefits and Risks of SGLT2 Inhibitors on the Renal System

Both substantial adverse effects and favorable outcomes for kidney disorders are associated with SGLT2 inhibitors. By suppressing the renin pathway through reduced sodium reabsorption, SGLT2 has an impact on the renin-angiotensin-aldosterone system (RAAS) pathway in the kidneys. By minimizing hyperfiltration damage, the downregulation of this mechanism provides renoprotection [11]. The RAA axis is upregulated in patients with diabetic kidney disease (DKD), which is responsible for many progressive kidney diseases. Patients with DKD have higher levels of angiotensin-2 (ANG2) and, as a result, higher levels of SGLT1/2 receptors [10,12]. The glucose excretion can control glucose levels in diabetics [13]. Reduced renal glucose reabsorption and increased insulin sensitivity are two benefits of SGLT2 inhibitors. By excreting glucose, these effects contribute to caloric loss and weight loss [13,14].

An SGLT2 inhibitor known as dapagliflozin has been shown to benefit individuals with CKD by slowing the course of the disease and reducing albuminuria [15]. Canagliflozin was shown in a different study, the CREDENCE trial, that reduces the chance of developing end-stage renal disease and having one’s creatinine level double by at least 32 percent [14].

One innovative application for SGLT2 inhibitors is helping patients who have just undergone kidney transplants. Although the results are limited, SGLT2 assist kidney transplant recipients in maintaining glycemic control, maintaining healthy body weight, and reducing uric acid levels (KTR) [15].

When prescribing SGLT2 inhibitors, one should also consider the potential adverse effects. According to the findings of one study, patients with T2DM who took SGLT2 inhibitors were at an increased risk for developing acute kidney injury (AKI), particularly when the drugs were combined with non-steroidal anti-inflammatory drugs (NSAIDs), anti-Ras, or diuretics [15]. There is evidence that dapagliflozin contributes to the advancement of renal dysfunction [11]. Another study has revealed the link between pyuria and urine microbiome dysbiosis, as well as other unfavorable outcomes, in diabetes patients treated with SGLT2 inhibitors [16].

Although SGLT2 inhibitors may have some unfavorable side effects, the overall performance of this class of drugs appears to be quite promising for many patients suffering from CKD, reduced Glomerular Kidney Filtration (GFR), and albuminuria.

### 3.2. Benefits and Risks of SGLT2 Inhibitors on the Hepatic System

Researchers have been looking into how SGLT2 inhibitors affect the liver to determine the extent of potential metabolic advantages. These drugs, in particular, have demonstrated low adverse events associated with other glycemic control medications, specifically hypoglycemia and diabetic ketoacidosis (DKA). SGLT2 inhibitors are effective because they block glucose reabsorption in the proximal convoluted tubule of the kidney, which allows glucose to be excreted in the urine. This mechanism operates independently of insulin and demonstrates the potential to reduce body mass index (BMI) and enhance CV outcomes [17]. Additionally, SGLT2 inhibitors have demonstrated promise in improving liver biomarkers and recovery from liver injury, hepatic fibrosis, and steatosis [18,19]. In most studies, therapy with SGLT2 inhibitors led to a statistically significant decrease in ALT and AST compared to a non-statistically significant decrease in the control group for other clinical parameters. In most investigations, γ-glutamyl transferase (GGT) and HbA1c decreased significantly after SGLT2 inhibitors therapy while they reduced non-significantly in the control group [18,19].

Non-alcoholic fatty liver disease is characterized by fat accumulation in the liver that results from factors other than alcohol consumption, drug use, or hypothyroidism. According to a recent thorough assessment of the literature, patients with T2DM who took various SGLT2 inhibitor medications had improved non-alcoholic fatty liver disease (NAFLD). It is unclear if the improved mechanism is caused directly by the SGLT2 inhibitor compounds or is mediated through metabolism [18,19]. On the other hand, in one of the studies mentioned above, it is good to know that the characteristics of Japanese and Caucasian patients differ significantly. For instance, the BMI of Japanese study participants is significantly lower than that of Caucasians, and Japanese pancreatic cells are significantly more susceptible to hyperglycemia than Caucasians. Indeed, under diabetes conditions, insulin secretory ability is quickly diminished in Japanese subjects [19]. Because of the changes brought on by insulin resistance and the changes in metabolic indicators, T2DM is linked to an increased risk of developing NAFLD. NAFLD is a primary cause of cryptogenic cirrhosis, which can proceed to cirrhosis, and 17% of T2DM patients with NAFLD had advanced liver fibrosis (78 million) [20].

Increased lipogenesis of liver is brought on in NAFLD patients by T2DM-induced high glucose levels [19]. The liver’s natural processes are interfered with by the proinflammatory adipokines generated as a result of this particular disease [21].

Exenatide and dapagliflozin together, as opposed to exenatide alone or with a placebo, improved liver enzymes and fatty liver indicators in T2DM patients, according to a recent clinical trial [21,22,23]. Combination therapy produced more weight loss in T2DM patients with NAFLD than either medication used independently or a placebo in these patients [21]. It has been demonstrated that a decreased BMI is associated with an increased NAFLD activity score [21].

SGLT2 inhibitors reduced controlled attenuation parameter and liver stiffness measurement in another meta-analysis despite considerable heterogeneity. Hence, SGLT2 inhibitors may delay hepatic fibrosis and steatosis and treat NAFLD specifically [22,23]. Therefore, further long-term, randomized, double-blinded, multi-centered clinical trials of SGLT2 inhibitors on hepatic fibrosis and steatosis are needed to help patients and physicians make the best treatment decisions.

Patients with T2DM who were given dapagliflozin for treatment for eight weeks experienced a reduction in liver fat and volume, according to the findings of a more recent randomised controlled trial. In addition, it has also come to light that lowering fibroblast growth factor 21 (FGF21) improves mitochondrial function [24]. This strategy lowered visceral adipose tissue (AT) and the inflammatory biomarker Interleukin-6 (IL-6) substantially [24]. The reduction in visceral AT results from the loss of liver fat and improvement in NALFD. High amounts of IL-6 in the blood are connected with myocardial infarctions, which is the importance of decreasing IL-6 [24].

A literature review looked for possible mitochondrial benefits of SGLT2 inhibitors managed to find that they can modulate mitochondrial functions through at least three different pathways: by changing the biogenesis and morphology of mitochondria, by controlling the generation of reactive oxygen species in mitochondria, and by regulating the amount of adenosine triphosphate (ATP) produced in mitochondria [25].

Based on these findings, SGLT2 inhibitors appear to be a compound with great potential for treating hyperglycemia and metabolic disorders.

### 3.3. Benefits and Risks of SGLT2 Inhibitors on the Pancreas

Although SGLT2 inhibitors’ main actions occur in the nephron, the fact that they are involved in controlling glucose allows us to consider the effects on the pancreas. Because these medications promote glycosuria, glucagon has an immediate effect on the pancreas [26].

The alpha pancreas cells contain the SGLT2 proteins on their own. These SGLT2 transporters are made more active by glucagon in alpha cells, but they do not colocalize with insulin or somatostatin in beta cells. This shows that when an excess of glucose has not been controlled, glucagon and SGLT2 proteins are directly related [27]. With the administration of SGLT2 inhibitors, there is an increase in glucagon because lower glucose absorption causes more gluconeogenesis and less glycolysis, which limits the production of ATP.

Dapagliflozin is the only SGLT2 inhibitor that demonstrates significant interactions with glucagon. Several studies have shown that dapagliflozin increases glucagon secretion and liver metabolism, especially gluconeogenesis, shortly after delivery, decreasing hypoglycemia episodes that can be hazardous or episodes where Empagliflozin is more efficient with its mechanism when given in conjunction with a DPP-4 inhibitor, such as linagliptin, and this further improves glycemic control in an insulin-resistant condition [28]. Since less insulin must be secreted due to the improved sensitivity, less nicotinamide adenine dinucleotide phosphate (NADPH), phosphatidylinositol 3′-kinase-dependent free radical generation results as an additional confusing effect of empagliflozin administration [28].

It is clear from looking at how SGLT2 inhibitors affect the pancreas that the mechanisms used to stimulate alpha and beta cells are different. According to research on the pancreatic side effects of SGLT2 inhibitors, notably dapagliflozin, pancreatitis happened intermittently in 1% of patients. However, these results can be the result of confounding factors considering that diabetes and hyperglycemia are already important risk factors for pancreatitis.

### 3.4. Benefits and Risks of SGLT2 Inhibitors on the Central Nervous System

Numerous brain regions, including the cerebellum, hippocampus, and blood–brain barrier (BBB), contain SGLT2 receptors. SGLT2 inhibitors are being intensively studied for their potential to prevent or protect against specific neurological diseases. SGLT2 inhibitors reduce reactive oxygen species (ROS), minimise BBB leakage, and decrease microglia load and acetylcholinesterase levels; these are the three primary pathways linking SGLT2 inhibitors to cognitive performance [29]. SGLT2 inhibitors are lipid-soluble and can pass the BBB, achieving a brain-to-serum ratio of 0.3 (Canagliflozin and Dapagliflozin) to 0.5 (Empagliflozin) [30]. There are SGLT receptors in the central nervous system (CNS). Many isoforms of these proteins can be identified in many regions of the CNS. Inhibitors of SGLT1 are found in pyramidal cells of the brain cortex, Purkinje cerebellum cells, pyramidal hippocampus cells, and granular cells [31]. SGLT2 expression in the brain is lower than that of SGLT1, and it occurs mainly in the microvessels of the blood–brain barrier, as well as in the amygdala, hypothalamus, periaqueductal gray, and dorsomedial medulla—the nucleus of the solitary tract [32].

Parkinson’s disease (PD) and Alzheimer’s disease (AD) are common age-related neurodegenerative conditions. AD also affects glucose metabolism; hence, it is frequently referred to as “Type 3 diabetes” or “diabetes of the brain” [33]. Patients with T2DM had a 53% greater relative risk of AD compared to non-diabetic individuals, according to a meta-analysis by Zhang J. et al. [34]. SGLT2 inhibitors may help Alzheimer’s patients through anti-inflammatory, anti-oxidative, or atheroprotective effects, as well as through direct neuroprotective effects like increasing brain-derived neurotrophic factor (BDNF) and blocking acetylcholinesterase (AChE). Insulin resistance is found in 8 out of 10 AD patients [35].

In prior investigations with mouse models, SGLT2 inhibitors therapy significantly decreased AD pathology, including tau phosphorylation and senile plaque density. This effect was related to enhancing cognitive functions, including memory and learning processes, as measured by the new object discrimination test and the Morris water maze test [36].

In the CNS, selective SGLT2 inhibitors have a place since, according to the findings of Erdogan MA. et al., Dapagliflozin lowers seizure activity at both the electrophysiological and clinical levels in a rat model of epilepsy [37]. No clinical data compare the effectiveness of ketogenic diets and dapagliflozin medication on brain epileptic activity; nonetheless, adherence to a ketogenic diet is challenging and must be regularly monitored. Contrarily, dapagliflozin is a safe medication routinely prescribed to diabetic patients. Cognitive impairment shares the same risk factors as epilepsy and atherosclerosis, and anti-epileptic medications, such as phenytoin, carbamazepine, and valproic acid, are associated with an elevated CV risk [38].

Lin B. et al. demonstrated an additional potential effect of SGLT2 inhibitors on the CNS for empagliflozin, which dramatically enhanced cerebral BDNF (brain-derived neurotrophic factor) levels in db/db mice. In addition, this effect was accompanied with enhanced cognitive functions [39].

Cerebrovascular dysfunction is a brain disorder associated with vascular pathology. A hyperglycemic state damages the microvascular structure of the brain, resulting in neurovascular remodeling, including a loss of endothelial integrity, basement membrane thickening, loss of myelin and neurons, and disruption of astrocytes and pericytes [40]. Empagliflozin demonstrated a neuroprotective impact on neurovascular remodeling in a mouse model of T2DM [41].

Ischemic or hemorrhagic blood flow disturbances are the most common causes of cerebrovascular dysfunction. Empagliflozin reduced neuronal mortality, infarct size, and cognitive impairment via HIF-1/VEGF signalling in a dose-dependent manner in a rat model of cerebral ischemia/reperfusion damage [42]. By avoiding neurovascular remodelling and lowering well-known risk factors for stroke, SGLT2 inhibitors may preserve cognitive functioning in diabetes individuals. Reducing inflammation, salt influx, and the HIF-1/VEGF pathway can also benefit individuals with post-stroke.

Due to the presence of SGLT2 in the CNS, SGLT2 inhibitors, which are used to treat diabetes and have other favourable metabolic effects, have been revealed to have potential neuroprotective qualities. These data, taken together, point to the potential therapeutic value of empagliflozin and dapagliflozin for various neurological disorders. New additional research is required in this relatively new research field.

### 3.5. Benefits and Risks of SGLT2 Inhibitors on the Pulmonary System

The prevalence of several respiratory diseases decreased with the use of SGLT2 inhibitors, including acute pulmonary edema, bronchitis, chronic obstructive pulmonary disease, asthma, non-small cell lung cancer, pleural effusion, pneumonia, pulmonary edema and masses, respiratory tract infections, and sleep apnea syndrome [43]. According to the findings presented in another study, pretreatment with empagliflozin was found to have powerful pulmonary protective effects against reperfusion-induced lung injury in vivo [43,44]. These benefits were related to the activation of pulmonary ERK1/2. In addition, suppression of ERK1/2 activation was sufficient to nullify the protective effects of empagliflozin on the lungs completely. The lung protection induced by empagliflozin and observed in the study is therefore associated with ERK1/2-dependent pathways. SGLT2 inhibitors may offer a potential and novel therapeutic option for patients at a high risk of sustaining lung injury during the perioperative phase. This is because SGLT2 inhibitors have considerable protective efficacies and a unique insulin-independent mode of action [44]. In addition, the risk of developing acute pulmonary oedema, asthma, and sleep apnea syndrome can all be significantly decreased by taking SGLT2 inhibitors [45].

According to another study’s findings, dapagliflozin could not improve the remodeling and dysfunction of the right ventricle in response to pressure overload with or without pulmonary angiopathy, nor could it reduce the amount of pulmonary vascular remodelling in rats with pulmonary arterial hypertension [46].

In a large, national cohort trial, the use of SGLT2 inhibitors was linked with decreased risks of total respiratory events, pneumonia, and respiratory failure among T2DM patients compared to the use of DPP-4 inhibitors [47]. All SGLT2 inhibitor compounds exhibited these respiratory improvements, indicating a possible class effect [47]. In another retrospective cohort analysis of individuals with T2DM in Hong Kong, SGLT2 inhibitor use was linked with a decreased incidence of incident obstructive airway disease (OAD) and a lower rate of OAD exacerbations in clinical settings compared to DPP-4 inhibitors use. Identical outcomes were likewise reported in both males and women [48]. According to the findings presented in another retrospective study, patients with T2DM on SGLT2 inhibitors have a lower incidence of pneumonia and sepsis, as well as a lower mortality risk associated with pneumonia, sepsis, and infectious diseases, in comparison to those initiated on DPP-4 inhibitors, regardless of age, sex, prior CV disease, or type of SGLT2 inhibitor used [49].

Further studies on SGLT2 inhibitors for primary and secondary prevention of respiratory illnesses are going to be done as a result of these results.

### 3.6. Benefits and Risks of SGLT2 Inhibitors on the Cardiovascular System

Inhibitors of sodium-glucose cotransporter 2 (SGLT2) protect myocardium by slowing the production of inflammatory chemokines [50]. The impact of SGLT2 inhibitors on fat loss is the primary contributor to this outcome. The death and hospitalisation rates of dapagliflozin and empagliflozin patients were evaluated in a recent meta-analysis of eleven CV outcomes trials involving 77,541 patients. The study found a reduction in mortality and hospitalisation rates regardless of T2DM status [51]. This claim proves their value as medicine in lowering the risk of death from CV problems. Blood pressure reduction (systolic and diastolic) is another CV advantage gained from SGLT2 inhibitor usage [52]. The association between SGLT2 inhibitors and a lower risk of nine different forms of CV disease was established using a meta-analysis of nine significant clinical trials. Conditions like atrial fibrillation, acute heart failure, bradycardia, hypertension, hypertensive emergency, and varicose veins were some of the CV disorders whose incidences were reduced [53]. An increase in hematocrit level is yet another effect attributed to SGLT2 inhibitors. The researchers investigating this effect hypothesised that it resulted from plasma volume contraction and diuresis [53].

Another study reveals that empagliflozin medication may improve anthropometric measurements, metabolic parameters, and blood pressure in T2DM with established coronary heart disease without impairing kidney function [54].

The use of SGLT2 inhibitors has been linked in several studies to improvements in lipid profiles, including higher HDL and lower LDL cholesterol and lower triglyceride level [52,53]. The ratio of HDL-C to LDL-C remained unchanged because the peaks in HDL and LDL were very similar [53,55]. Another set of findings from a study demonstrated that empagliflozin, primarily acting on SGLT2, prevented DNA methylation changes brought on by high glucose [56]. These findings also provided evidence of a new mechanism by which SGLT2 inhibitors can exert cardio-beneficial effects [56]. Dapagliflozin is safe and improves outcomes regardless of baseline NT-proBNP concentrations in HF with mildly reduced Ejection Fraction (HFmrEF) or HF with preserved Ejection Fraction (HFpEF), with the greatest absolute benefit likely seen in patients with higher NT-proBNP concentrations according to the findings of another large trial [57]. One of the other effects of their use that have been documented is a decrease in uric acid levels [52,53,55,56]. Empagliflozin reduces unfavourable cardiac remodelling in HF by increasing the switch of myocardial fuel usage from glucose to ketone bodies, enhancing myocardial energetics, systolic function, and cardiac reversal remodelling [58]. Because SGLT2 inhibitors are weak diuretics, volume depletion-related side effects such as hypotension, syncope, and dehydration are more likely to occur [59]. Before starting SGLT2 inhibitors therapies, the patient’s volume status should be assessed. In cases of low volume status, for instance, dose changes for loop diuretics may be required.

The effects of this drug class have been verified by several relevant studies, and they have the potential to be associated with a lower risk of CV diseases.

## 4. Discussion

The EMPA-REG OUTCOME was the first study to suggest an effect of SGLT2 inhibitors on CV outcomes. An amount of 7028 diabetic patients (BMI 45 kg/mq, eGFR > 30 mL/min/1.73 mq, and HbA1c between 7 and 10) were randomised to receive standard or high-dose empagliflozin (respectively, 10 mg or 25 mg) or placebo and observed for a median observation time of 3.1 years [58]. The study demonstrated the efficacy of empagliflozin in reducing deaths from CV causes, hospitalisations for HF, and deaths from any cause compared to a placebo, regardless of the administered dose. During this time, the EMPA-REG OUTCOME trial showed that the use of empagliflozin resulted in a reduction of macroalbuminuria that was up to 55% lower than that of the placebo [58,60].

A few years later, the EMPEROR-Reduced trial with a randomised group of 3730 patients (NYHA classes II to IV, eGFR > 20 mL/min/1.73 mq and ejection fraction (EF) below 40%) to receive empagliflozin (10 mg once daily) or placebo to assess the efficacy of empagliflozin in HFrEF, regardless of diabetes, was conducted [61]. The results showed that empagliflozin helped lower the number of hospitalisations for HF (*p* = 0.001) and improved quality of life. It was much more effective in patients with lower EF. Also, the estimated GFR decreased more slowly in the empagliflozin group (*p* = 0.001), suggesting that SGLT2 inhibitors may have a protective impact on the kidneys [60]. Specifically, the DAPA-HF trial demonstrated that dapagliflozin reduced by 26% the relative risk of HF hospitalisation/urgent visits for HF and CV mortality in HFrEF (patients with NYHA classifications II-IV and eGFR > 30 mL/min/1.73 mq) [61].

Patients enrolled in the EMPEROR-Preserved trial had more comorbidities, lower EF, and higher NT-proBNP levels than those in other HFpEF studies, such as the PARAGON-HF study [62,63,64].

The importance of SGLT2 inhibitors was further reinforced by a meta-analysis of the most major trials on gliflozins and HF (DELIVER, EMPEROR-Preserved, EMPEROR-Reduced, DAPA-HF, and SOLIST-WHF), regardless of EF, supporting their function as a cornerstone in HF therapy [65]. All of the aforementioned trials examined the effects of gliflozin on stable, ambulatory patients. By contrast, a recently conducted trial called EMPA-RESPONSE-AHF, which was later validated by the EMPULSE trial, randomly assigned patients hospitalised with HF to receive empagliflozin or a placebo within 24 h of admission [66]. Without regard to left ventricular EF (LVEF), 530 patients with newly diagnosed or worsening chronic HF were randomly assigned to receive either empagliflozin 10 mg once daily or a placebo. Those who received the drug saw reduced all-cause mortality and improved quality of life (*p* = 0.0054) [66].

To provide HF patients with the best possible medical care in the shortest amount of time, the European Heart Failure Working Group recently published a document recommending empagliflozin as first-line therapy as soon as possible. In patients with HF, gliflozins have demonstrated undeniable clinical advantages. However, gliflozins’ methods of action are still up for debate. Furthermore, SGLT2 inhibitors do not require up-titration, which is a major benefit on top of the already-mentioned positive advantages. Additionally, gliflozins are exceptionally well tolerated due to the low frequency and severity of their side effects among patients using SGLT2 inhibitors, as we can see in Table 2.

SGLT2 inhibitor medicines are effective and safe, with only minor and tolerable side effects.

Fungal infections of the urogenital tract are the most commonly reported adverse effect [59,67]. These infections are more common in women and the elderly. A bacterial infection in the urinary tract has the potential to develop into pyelonephritis or urosepsis. Hypoglycemia, excessive volume depletion, Fournier’s gangrene, and ketoacidosis are all possible, though uncommon, side effects of gliflozins. It is also unclear whether or not they affect cholesterol levels or whether they favour bone fracture risk [68,69,70]. There has been talking about a spike in LDL cholesterol levels, but whether or not this is due to gliflozins is still up for debate after trials produced conflicting results [52,67]. However, there was clear evidence of increased HDL levels among these patients [59,71].

Over 2500 cases of DKA were reported to the FDA between 2014 and 2016; these cases were more common in insulin-treated individuals and occurred even at glucose blood levels below 250 mg/dl [68,69,70]. The 2020 FDA advisory warnings [72] for increased risk of perioperative euglycemic DKA say that diabetic patients should stop taking canagliflozin, dapagliflozin, empagliflozin, and ertugliflozin at least three days and four days before surgery. Even though the best time to stop taking these drugs before surgery in HF patients is still debated, there is no evidence to show when this should be done. In these patients, stopping the drug could hurt how HF is treated [73].

Type 1 diabetes and an eGFR of less than 25 mL/min per 1.73 m^2^ are contraindications for SGLT2 inhibitors. When beginning SGLT2 inhibitor in patients with volume depletion, active vaginal mycotic infections, hypotension < 95 mmHg, and diabetic ketoacidosis, caution should be exercised [74,75]. Patients should be cautioned against taking SGLT2 inhibitors during illness and if they cannot maintain enough fluid intake or have an acute renal injury.

In addition to the landmark trials, barriers to prescribing SGLT2 inhibitors remain an essential discussion area. Despite the excellent benefits of SGLT2 inhibitors and updated clinical guidelines, SGLT2 inhibitor prescriptions for eligible patients remain limited [76]. These include a misunderstanding of side effects and a deficiency of guidelines for beginning SGLT2 inhibitors in elderly individuals [77]. In addition, cost and insurance restrictions impede patients from gaining access to these costly, effective, and protective drugs.

Future studies will continue to investigate the application of SGLT2 inhibitors in type 1 diabetes and pre-diabetes. There are currently few investigations on the use of SGLT2 inhibitors in type 1 diabetes [78]. Current research continues to investigate the safety of SGLT2 inhibitors in patients with an eGFR of less than 30 mL/min per 1.73 m^2^. Upcoming clinical trials can further investigate SGLT2 inhibitors’ capacity to reduce blood pressure. Now, the EMPACT-MI trial is exploring the use of empagliflozin in patients who have experienced an acute myocardial infarction, as well as the prospect of reducing the risk of HF and death.

Flozins may have beneficial effects on T2DM, atherosclerosis, and cognitive impairment via multiple mechanisms. However, long-term clinical trials are required to determine whether the aforementioned pathways are clinically meaningful, as the atheroprotective and neuroprotective effects of SGLT2 inhibitors are not instantaneous and require long-term administration.

As previously stated, SGLT2 inhibitors substantially lower CV risk. By lowering vascular inflammation, oxidative stress, and endothelial dysfunction, they produce a pleiotropic anti-atherosclerotic action [79]. In a prior trial, including diabetic individuals, a three-month therapy with empagliflozin reduced complex intima media thickness (cIMT) by 7.9%. Remarkably, this benefit was already significant after just one month of empagliflozin medication [80]. CIMT is commonly tested in the carotid arteries [81], as it is a significant indication of early atherosclerosis. I. Feinkohl et al. [82,83] report that cIMT is also a significant predictor of cognitive decline in T2DM patients. Future research should assess the therapeutic significance of SGLT2 inhibitors’ capacity to minimise atherosclerotic lesions and the consequent impact on cognitive functions.

Gliflozins can potentially improve renal function, glucose blood levels, cardiac function, and remodelling, making them the ideal medicine to stop the circles from progressing or to decrease their progression. All of the benefits of SGLT2 inhibitors can be shown in Figure 1.

In this review, our primary objective was to evaluate the efficacy of SGLT2 inhibitors in terms of their positive and negative consequences on the CV, pulmonary, nervous, hepatic, and renal systems and the pancreas in diabetic patients. There were a strikingly greater more significant number of protective factors than risk factors, which is evidence of the high and widespread application of these medications, which are thought to benefit individuals with hyperglycemia. These positive outcomes come at the expense of a medicine that is generally well-tolerated, has few adverse effects, and does not require titration. Further study will aid in comprehending the fantastic benefits of SGLT2 inhibitor drugs.

## 5. Conclusions

This narrative review highlights the efficacy and widespread use of this class of medications by many protective factors in hyperglycemic people. Weight loss, enhanced liver enzymes and biomarkers, optimum glucose homeostasis attributable to the pancreatic hormones insulin and glucagon, reduced risk of cardiovascular and neurological illnesses, cardioprotective effects, and lowered blood pressure are only some of the advantages. On the other hand, volume depletion and hypotension, pancreatitis, and episodes of transient hypoglycemia are some of the adverse consequences in a tiny percentage of those using SGLT2 inhibitors. These complications can damage the patient, but they only happen about 1% of the time and may result from chance or the patient’s underlying health condition.

This literature study led us to conclude that SGLT2 inhibitors are helpful for the vast majority of patients, particularly those with renal impairment. We are hopeful for the further development of this medication class to minimize the potential risks.

## Figures and Tables

**Figure 1 medicina-59-00742-f001:**
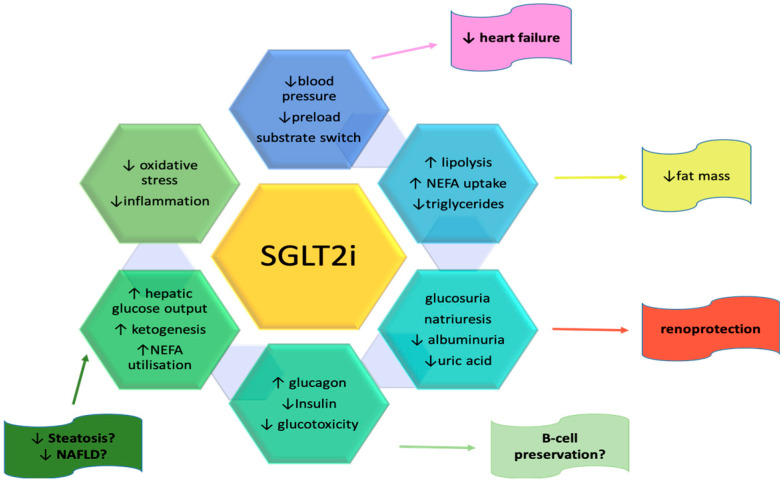
Benefits of SGLT2 inhibitors. (“↑”: increase; “↓”: decrease; SGLT2i: sodium-glucose cotransporter 2 inhibitors; NAFLD: non-alcoholic fatty liver disease; NEFA: nonesterified fatty acids) [84].

**Table 1 medicina-59-00742-t001:** Pharmacological characteristics of SGLT2 inhibitors available in the market.

SGLT2 Inhibitors	Therapeutic Indication	Dosing
Canagliflozin	Type 2 diabetes mellitus	100 mg once a day
Dapagliflozin	Type 2 diabetes mellitus	10 mg once a day
	Heart failure	10 mg once a day
	Chronic kidney disease	10 mg once a day
Empagliflozin	Type 2 diabetes mellitus	10 mg once a day
	Heart failure	10 mg once a day
Ertugliflozin	Type 2 diabetes mellitus	5 mg or 10 mg once a day
Sotagliflozin	Type 1 diabetes mellitus	200 mg or 400 mg once a day
Bexagliflozin	Type 2 diabetes mellitus	20 mg once a day

**Table 2 medicina-59-00742-t002:** Adverse events of SGLT2 inhibitors.

	Canagliflozin	Dapagliflozin	Empagliflozin	Ertugliflozin	Sotagliflozin	Bexagliflozin
Hypotension	uncommon	uncommon	Very common	common	common	unknown
Diabetic ketoacidosis	rare	rare	uncommon	rare	common	-
Bone fracture	uncommon	-	-	-	-	-
Genital infections	common	common	common	common	common	common
Urinary tract infections	common	common	common	common	common	common
Fournier’s gangrene	unknown	very rare	rare	unknown	rare	unknown
Amputation of lower limbs	uncommon	unknown	unknown	unknown	unknown	rare
Hypoglycaemia	very common in combination	very common	very common	common	-	-

## Data Availability

Not applicable.

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
