# Peer review of "Sodium-Glucose Cotransporter 2 (SGLT2) Inhibitors: Harms or Unexpected Benefits?"

_medicina, 2023, doi:10.3390/medicina59040742_

Round 1

Reviewer 1 Report

Dears:

Receive a cordial greetings,

It is an interesting work that is promoted as a contribution to know the advances on the benefits or disadvantages of sodium-glucose cotransporter 2 (SGLT2) inhibitors, I only have the following suggestions in the introduction: adding the following reference could enrich a little more than the initial context:

Martínez, M. S., Manzano, A., Olivar, L. C., Nava, M., Salazar, J., D’marco, L., Ortiz, R., Chacín, M., Guerrero-Wyss, M., Cabrera de Bravo, M., Cano, C., Bermúdez, V., & Angarita, L. (2021). The role of the α cell in the pathogenesis of diabetes: A world beyond the mirror. International Journal of Molecular Sciences22(17), [9504]. https://doi.org/10.3390/ijms22179504

On the other hand, I suggest small changes or minimal improvements in the language, and because the time range for the expert search has been wide, I could suggest if possible adding at least 2 or 3 references to the year 2022 or 2023 alluding to the main axis of their theme, if they exist.

Author Response

Dear Reviewer 1,

Thank you for your recommendations.

In the introduction, we have added the suggested reference that is very appropriate for our manuscript and revealed an important detail about chronic hyperglycemia. The language revision was made using a specialised program. We would like to mention that our manuscript already had references from the 2022 year, but we added a new reference from the 2023 year.

Reviewer 2 Report

Abstract

Line 11-12: diabetes is a metabolic disease. Please edit for accuracy. Same for line 23-24.

Introduction

Line 30: Please do not refer to the transporters and the inhibitors of those transported by the same abbreviation. Perhaps SGLT2 for the transporters and SGLT2i for the inhibitors.

Section 3.2

Lines 111 to 112: Redundant, remove.

This section requires more information. The first part rehashes what has already been said earlier. Perhaps provide a bit more detail based on ref 17-18, which were cited on lines 121 to 122.

Lines 140 to 142: please link this statement to SGLT2i and reduction in BMI

I do not see any adverse effects discussed.

Section 3.3

Lines 157 to 158: This statement requires a valid reference as it goes against what is known. SGLT2i by themselves do not cause hypoglycemia.

Lines 167 to 169: more appropriate for the cardiovascular section

Section 3.4

please ensure proper use of paragraphs to organize thoughts. The section needs to be reorganized.

Lines 189 to 190: I suspect the authors wanted to say “…, contain SGLT2 receptors” instead of “…, contain SGLT2 inhibitors.”

Lines 202 to 203: this statement requires a reference. If the reference is [26], please re-write the paragraph to properly situate the reference and text.

Lines 234 to 236: state a source for this statement. If your systematic search yielded no results, state so.

Section 3.5

Again, proper use of paragraphs is needed.  

No adverse effects discussed.

Section 3.6

No adverse effects discussed.

Discussion

Appears to be a discussion on the CV benefits of SGLT2i. Needs to be a lot more representative of the stated objectives of the study.

Table 2 needs a reference(s).

Line 365, 373, “efficacy” is not the proper term to use for the work done here. Please revise.

Author Response

Dear Reviewer 2,

Thank you for your recommendations. Each recommendation was analyzed and treated with interest in modifying the article’s structure. The language revision was made using a specialized program.

We have edited lines 11-12 and 23-24.

Introduction

-we corrected the misunderstanding between SGLT2 inhibitors and SGLT2 transporters

Section 3.2

-we removed lines 111-112

-we provided more details according to your recommendation; we detailed the references 17-18 and added new references of benefits of SGLT2 inhibitors on the hepatic system

-we also linked the statement to SGLT2 inhibitors and reduction in BMI and detailed it.

Section 3.3

Inhibition of SGLT2 results in glucose excretion in the urine, lowering blood glucose levels. Many clinical trials revealed that patients receiving SGLT2 inhibitors drugs have lower glycemia, body weight, and blood pressure than those taking other antidiabetic agents. The unique expression of SGLT2 by pancreatic alpha cells, where glucagon secretion is directly regulated, explained these observations. We removed lines 167-169

Section 3.4

We reorganized the paragraphs in this section and added new references. We explained more accurately where SGLT receptors are in the central nervous system and how these drugs can improve various neurological disorders. The cited articles were treated more carefully.

Section 3.5

We added new paragraphs and also new references. The cited articles were treated more carefully. Also, we would like to mention that adverse effects are mentioned in the discussion topics.

Section 3.6

We added adverse effects of SGLT inhibitors on the cardiovascular system, such as hypotension, syncope, and dehydration.

Discussion:

Significant SGLT2 inhibitors clinical trials have repeatedly demonstrated cardiovascular and renal benefits. In recent years, it has become evident that SGLT2 inhibitor is not only a diabetic drug, but also a cardiovascular and renal disease-modifying agent; that’s why we focused on discussing the most representative trials in this topic.

We added the contraindications for SGLT2 inhibitors.

In this section of our manuscript, we have reported the adverse reactions of SGLT2 inhibitors. We added new paragraphs to be more representative of the stated objectives of our study. Also, we mentioned that future research should assess the therapeutic significance of SGLT2 inhibitors on other systems, other than cardiovascular, renal, and diabetes mellitus (where are the significant trials).

Reviewer 3 Report

Dear Authors

I carefully read and checked the article. This article is a review about SGLT2 inhibitors. I saw a few typos, SGLT2 inhibitors are misspelled as SGTL2. The parts that I think are wrong are highlighted in yellow and the correct one is indicated in the text box. Introduction is adequate, material method is neat, this article is a review on the benefits and risks/side effects of sglt-2 inhibitors. For this purpose, the benefits and harms of sglt2s are explained in detail as a separate paragraph for each organ. English is very understandable, I enjoyed reading it. SGLT-2 inhibitors are important drugs that have emerged in the treatment of type-2 diabetes in recent years and provide especially cardiac and renal protection, and the current literature has been reviewed and important information has been given. Side effects are also mentioned. However, it is more common than euglycemic diabetic ketoacidosis, which is a rare but important side effect.

 Therefore, i recommend a few articles from current literature.

Article-1: Journal of the College of Physicians and Surgeons--Pakistan: JCPSP, 32(7), 928-930.

Article-2:  Ir J Med Sci 191, 1647–1652 (2022).

Author Response

Dear Reviewer 3,

Thank you for your recommendations.

The misspelled words as „SGTL2” were corrected.

 Thank you for recommending some new references, one of which we used in our manuscript.